# Endogenous Pain Modulation in Response to a Single Session of Percutaneous Electrolysis in Healthy Population: A Double-Blinded Randomized Clinical Trial

**DOI:** 10.3390/jcm11102889

**Published:** 2022-05-20

**Authors:** Sergio Varela-Rodríguez, José Luis Sánchez-Sánchez, Enrique Velasco, Miguel Delicado-Miralles, Juan Luis Sánchez-González

**Affiliations:** 1Department of Nursing and Physiotherapy, Faculty of Nursing and Physiotherapy, University of Salamanca, 37007 Salamanca, Spain; jlsanchez@usal.es (J.L.S.-S.); juanluissanchez@usal.es (J.L.S.-G.); 2Instituto de Neurociencias de Alicante (CSIC-UMH), 03550 Alicante, Spain; e.velasco@umh.es (E.V.); mdelicado@umh.es (M.D.-M.)

**Keywords:** percutaneous electrolysis, electrical stimulation, pain management, pain threshold, postsynaptic potential summation, randomized controlled trial

## Abstract

The purpose of this double-blinded randomized controlled trial was to investigate whether percutaneous electrolysis (PE) is able to activate endogenous pain modulation and whether its effects are dependent on the dosage of the galvanic current. A total of 54 asymptomatic subjects aged 18–40 years were randomized into three groups, receiving a single ultrasound-guided PE intervention that consisted of a needle insertion on the lateral epicondyle tendon: sham (without electrical current), low-intensity (0.3 mA, 90 s), and high-intensity (three pulses of 3 mA, 3 s). Widespread pressure pain thresholds (PPT), conditioned pain modulation (CPM), and temporal summation (TS) were assessed in the elbow, shoulder, and leg before and immediately after the intervention. Both high and low intensity PE protocols produced an increase in PPT in the shoulder compared to sham (*p* = 0.031 and *p* = 0.027). The sham group presented a significant decrease in the CPM (*p* = 0.006), and this finding was prevented in PE groups (*p* = 0.043 and *p* = 0.025). In addition, high-intensity PE decreased TS respect to sham in the elbow (*p* = 0.047) and both PE groups reduced TS in the leg (*p* = 0.036 and *p* = 0.020) without significant differences compared to sham (*p* = 0.512). Consequently, a single PE intervention modulated pain processing in local and widespread areas, implying an endogenous pain modulation. The pain processing effect was independent of the dosage administrated.

## 1. Introduction

Percutaneous electrolysis (PE) is an ultrasound-guided minimally invasive technique consisting of the application of a galvanic electrolytic current through an acupuncture needle placed into affected soft tissue [1,2,3]. The objective of this therapy is proposed to generate an analgesic effect and reactivate the local inflammatory response that allows phagocytosis and repair in the region of treatment [2,4,5].

Although scientific evidence concerning PE is still limited compared to other physical therapies, the number of publications investigating the effectiveness of this treatment is progressively growing [6,7]. The efficacy of this technique has been mainly studied in musculoskeletal disorders; in fact, a recent meta-analysis has found moderate evidence suggesting a positive effect of PE for reducing pain and related disability in these pathologies [8]. Some of the conditions that could benefit from PE treatment are patellar tendinopathy, lateral epicondylalgia, shoulder pain, whiplash syndrome, or temporomandibular pain [4,6,9,10,11,12,13]. However, which musculoskeletal pain conditions would receive the greatest benefit from this approach and the ideal dosage are questions still unanswered [8].

Regarding the underlying mechanisms of this technique, they are not completely defined, and both mechanical and biochemical effects are currently suggested [5,8,9,14]. Firstly, it is proposed that the mechanical stimulation produced by the needle would induce remodeling of the extracellular matrix following fibrocyte activation [2]. Secondly, the electrical stimulation provided by the galvanic current in a saline solution would generate a non-thermal electrolytic reaction that promotes a controlled inflammatory response and wound healing in damaged and/or degenerated tissue [4,8,15,16,17]. Although these hypotheses have been proposed based on clinical results, and mechanistic research contrasting them is scarce, pioneering works have examined the histologic and molecular response produced by the PE application to tendon or muscle in rats. These authors have reported an increase in the expression of anti-inflammatory proteins, angiogenic factors, and some genes related to collagen regeneration and a decrease in pro-inflammatory mediators [5,18,19].

Recently, a third potential mechanism has been proposed: the neurophysiological effect [8]. This hypothesis is integrated into a pain neuroscience paradigm and has been suggested for other needling techniques such as dry needling, acupuncture, or electro-acupuncture [20,21,22]. The neurophysiological effect of these therapies might be produced by the activation of descending pathways, stimulation of the neuroendocrine system, conditioned pain modulation, or segmental inhibition [20,21,22,23,24]. These potential mechanisms should be included in the concept of endogenous pain modulation (EPM), which is the ability of the central nervous system to modulate nociceptive input from peripheral tissues as it ascends to the spinal cord/brainstem and the brain [25]. Thus, although the intensity of the nociceptive input is related with pain, the subsequent modulation of peripheral input conditions the intensity of the resulting perceptions [26,27].

The neurophysiological effect of PE has not been extensively studied. Some clinical trials reported a greater parasympathetic activity (detected by hearth-rate variability) during the application of PE in healthy humans [14,28,29]. De-Miguel-Valtierra et al. [9] examined the long-term effects on widespread sensitivity of patients with shoulder pain and found no significant differences between the application or not of the therapy. However, to the best of our knowledge, no studies have investigated the short-term effects of PE on EPM.

Therefore, the primary aim of this randomized clinical trial was to investigate if a single session of percutaneous electrolysis activates endogenous pain modulation mechanisms compared to a sham intervention in asymptomatic healthy participants. The secondary aim was to determine if these potential effects on EPM are different between the application of two different protocols of PE (low intensity galvanic current during longer time or high intensity during shorter time). Thus, we first hypothesize that a single session of percutaneous electrolysis will be able to activate EPM in a greater manner than the application of a sham intervention. Our second hypothesis is that activation of EPM will be higher with the high-intensity protocol than with the low-intensity protocol.

## 2. Materials and Methods

### 2.1. Study Design

This study was a double-blind randomized controlled trial. The development of this project took place at the Faculty of Nursing and Physiotherapy of the University of Salamanca (Spain). The protocol of the clinical trial received approval from the Ethics Committee of University of Salamanca (record number 550/2021) and was carried out in accordance with the Declaration of Helsinki.

The reporting of this study was conducted according to the CONSORT 2010 Statement (Consolidated Standards of Reporting Trials). The clinical trial was registered in ClinicalTrials.gov with the registration number NCT05097937.

### 2.2. Participants Recruitment and Eligibility Criteria

Asymptomatic volunteers were recruited from the University of Salamanca through social media and e-mail between November 2021 and February 2022. The inclusion criteria consisted of: (a) age between 18 and 40 years; (b) both genders; and (c) Spanish native speaker. The exclusion criteria were: (a) belonephobia or fear of needles; (b) previous or current lateral epicondylalgia; (c) cognitive and sensory disorders; (d) any medical condition causing pain; (e) cutaneous alterations; (f) pregnancy; (g) neurological, cardiovascular or metabolic diseases; (h) fibromyalgia; (i) frequent or recent (24 h before) consumption of alcohol and other drugs; (j) medical or physical therapy treatment in the last week; (k) intake of caffeine in the two h prior to measurement; and (l) vigorous physical activity on the day of testing.

All individuals were given an explanation of the study procedures, and those who accepted the requirements and met the eligibility criteria were included. Participants signed a written informed consent before any data were collected.

### 2.3. Allocation and Randomization

Once the informed consent was signed, subjects were randomly assigned into one of three interventions groups: (1) sham; (2) low-intensity; and (3) high-intensity. Concealed allocation was conducted with a computer-generated randomized table of numbers in a blocked proportion of 1:1:1 (GraphPad Software Inc., San Diego, CA, USA). A researcher, not involved in any other aspect of the experiment, carried out the randomization procedure. Individual and sequentially numbered index cards with the random assignment were folded in sealed opaque envelopes. The clinician opened the appropriate envelope after baseline data collection.

### 2.4. Blinding

Both participants and assessors were blinded to the group assignment. The only person aware of the intervention nature was the clinician carrying out the treatment, who was not involved in any other aspect of the research, such as analysis or data collection. In addition, the statistical analysis was conducted by an independent researcher not implicated in other aspects of the study.

### 2.5. Interventions

This randomized clinical trial had three groups: sham, low intensity, and high intensity. All groups received a single session of an ultrasound-guided needle intervention on the common extensor tendon of the lateral epicondyle of the right side. The intervention was performed by a physical therapist with more than 10 years of experience in the technique. Participants were lying comfortably in a supine position with the right elbow on the table in a position of 20° flexion and with the forearm pronated. As described by Rodríguez-Huguet [13], a 0.3 × 25 mm acupuncture needle (Agupunt, Barcelona, Spain) was inserted under ultrasound guidance at 45° to the skin in the direction of the lateral epicondyle to reach the deep surface of the common extensor tendon (Figure 1).

The needle insertion was maintained throughout the intervention duration (90 s) in all groups in order to increase the comparability between them. The three procedures only differed in the application time and the intensity of the galvanic current through the needle. Furthermore, the total electric charge applied in the low-intensity and high-intensity groups were equalized at 27 mC.

In all groups, the handle of the certified medical device used to apply PE (EPI^®^-Alpha, Barcelona, Spain) was connected to the needle, used as an electrode (cathode), so that the participant would always expect to receive the electrical current and masking would be preserved. An adhesive electrode was placed on the distal third of the right arm, which acted as an anode. Additionally, the electro-stimulator was customized in all groups to emit a sound indicating that the treatment has started.

The sham group received the needling insertion without the electrical current application (0 mA). This sham procedure was previously used by Fernández-Rodríguez et al. [30] in a randomized clinical trial investigating the effects of PE in patients with plantar heel pain.

The low-intensity group received a single pulse of galvanic current with an intensity of 0.3 mA for 90 s. Similar current parameters were used by de-Miguel-Valtierra et al. [9] and Arias-Buría et al. [10] exploring the effects of PE in patients with shoulder pain.

The high-intensity group had the needle inserted without current for 75 s. After this time, three galvanic current pulses were applied with an intensity of 3 mA and a duration of 3 s each one, with 3 s of rest between them, resulting in a 90 s intervention duration. This PE modality was employed by López-Martos et al. [11] and Moreno et al. [3] studying the PE effects (Figure 2A).

### 2.6. Outcomes

All outcomes were evaluated before (2 min) and immediately after (2 min) the intervention by a blinded assessor. The experiment always took place in the same room with a similar temperature and only in the morning due to the way the circadian rhythm influences the pain systems, especially the descending pain modulatory system [31]. Before the baseline assessment of the outcome variables, the socio-demographic variables were evaluated by anamnesis.

The outcome variables were the changes in widespread Pressure Pain Thresholds (PPT), Conditioned Pain Modulation (CPM), and Temporal Summation (TS). All these outcomes were assessed in that order at baseline and post-intervention using a digital pressure algometer (Force One FDIX, Wagner Instruments, Greenwich, CT, USA) with a round rubber surface of 1 cm^2^. Participants were in the same position described before in the intervention. Each outcome variable was assessed by algometry at three locations (on the right side) in the same order: common extensor tendon of the lateral epicondyle (treatment area), tibialis anterior muscle (a distant non-related area) and bicipital groove (a segmental-related area). These locations were marked with a pen to assess all measurements at the same points.

The remaining intervals between different measures were also planned so that the measurements would not be influenced by each other.

#### 2.6.1. Primary Outcome

The primary outcome was widespread PPT. The pressure was applied perpendicularly over the three points described above at a rate of approximately 1 kg/cm^2^ per second until the subject referred the slightest pain perception [32,33]. At this moment, the algometer was removed and the pressure applicated was recorded. Participants had two practice trials on the non-testing forearm before the evaluation began.

Two measurements were taken at each assessed point with a 30-s interval for avoiding temporal summation [34], and the average was used for the analysis. If there was a difference of more than 1 kg/cm^2^ between the two measurements at the same location, a third measurement was performed. The final value was determined after excluding the most extreme measurement and calculating the average of the other two.

#### 2.6.2. Secondary Outcomes

The secondary outcomes included CPM and TS of mechanical stimuli. The CPM assessment started one minute after the PPT measurement. First, the PPT scores already assessed were taken as non-conditioned stimuli. The conditioning stimulus consisted of an ischemic pain produced by a pressure cuff placed on the left arm. The pressure cuff was inflated through successive 10–20 mmHg insufflations until the subject felt the first sensation of pain [33]. This pressure was maintained for 30 s and then was adjusted to evoke a pain intensity perceived by the participant of 4/10 [35]. At that time, the PPTs were measured at the three marked points (conditioned PPTs). CPM score was calculated as the difference between conditioned and non-conditioned PPT scores. We have accomplished the recommendations of CPM testing [36].

The TS evaluation began one minute after the CPM assessment. The evaluation consisted of 10 mechanical pressure stimulus applied sequentially with the algometer at non-conditioned PPT level (with 1-s interval between them) in each of the three assessed points [37]. Subjects rated the pain intensity evoked by the 1st and 10th pressure on a verbally administered numerical pain rating scale (NPRS; 0 = no pain, 10 = maximum tolerable pain). The TS score resulted from the difference between the 10th and the 1st pain intensity rating scores [38]. The outcomes assessment is illustrated in Figure 2.

### 2.7. Sample Size Calculation

The proposed methodology has not yet been undertaken in any study involving PE. Therefore, we calculated the sample size based on the results reported by other papers that measured the same outcomes and used similar interventions [33,39]. We estimated an effect size of 0.47 (Cohen’s *d*) for between-groups differences in PPT (One-Way ANOVA), with a statistical power of 80% and an alpha error of 0.05. The estimated sample size was 16 subjects per group, but we added 10% due to possible dropouts, resulting in a recruited sample of 18 participants per group (54 subjects in total). The GPower 3.1 program (Düsseldorf, Germany) was used to calculate the sample size [40].

### 2.8. Statistical Analysis

Data were analyzed by a blinded independent statistician using the IBM-SPSS software package (IBM Corp. Released 2019. IBM SPSS Statistics for Windows, Version 26.0. Armonk, NY, USA: IBM Corp.). The significance level was established at 0.05 and the limits of the confidence interval at 95%. Changes in outcomes were tested on the basis of the intention-to-treat principle.

Firstly, we checked the assumption of normality for all variables with the Shapiro–Wilk test and plotting the data distributions. Then, a descriptive analysis of baseline characteristics of the sample was reported, calculating the mean and standard deviation for continuous variables or the absolute number and relative frequency percentage for categorical variables. 

Subsequently, to assess the between-groups effects for continuous variables, we conducted the analysis of variance—ANOVA—(or Kruskal–Wallis test, depending on the normality) and the Bonferroni test (or Mann–Whitney U tests) as a post hoc comparison over the pre vs. post differences observed in each group. The Chi-squared test was used to compare categorical variables between groups. Additionally, we explored the within-group effects (pre vs. post intervention) performing Student’s *t*-test for dependent samples (or the Wilcoxon test in case of non-normal distribution). The effect size (Cohen’s *d*) was calculated for the main variables and was classified as small (0.20 to 0.49), medium (0.50 to 0.79), or large (>0.80).

## 3. Results

### 3.1. Sample Characteristics

A total of 54 young and healthy participants were included and randomly allocated into three groups. All volunteers received the assigned intervention and completed the outcome assessments. Hence, there were no dropouts during the study nor any adverse effects during the interventions (Figure 3).

Characteristics of the participants are summarized in Table 1. No statistically significant pre-intervention differences in sociodemographic and outcome data were present between the groups. For text clarity, an in-deep representation of the statistical results is presented in Table 2, where effect sizes are also reported.

### 3.2. Widespread Sensitivity

Both low- and high-intensity groups increased PPT in the shoulder (*p* = 0.006 and *p* = 0.001, respectively) without differences in the elbow and leg (*p* > 0.05). In the sham group, there were no observable differences in any of the locations assessed (*p* > 0.05) (Appendix A).

Regarding the comparisons between groups, significant differences were found for the shoulder (*p* = 0.041) but not in the elbow or leg assessments (*p* = 0.161 and *p* = 0.761). Post hoc comparisons detected a significant increase in the shoulder PPT in low- and high-intensity groups compared to sham (*p* = 0.031 and *p* = 0.027, respectively) (Figure 4A). These results suggest that the application of galvanic current could be affecting pressure pain thresholds, producing a moderate hypoalgesia at a point close from the intervention site independent of its intensity of application, without affecting the treatment area (elbow) or a distant non-related area (tibialis anterior muscle).

### 3.3. Conditioned Pain Modulation

The sham group was the only intervention that decreased the CPM until returning an abolished CPM in average in the shoulder (*p* = 0.006, 0.00 ± 0.26 kg, Table 2). However, no intervention produced CPM with no changes at the elbow and leg (*p* > 0.05) (Appendix A). 

In the same direction, we found significant differences for CPM in the shoulder (*p* = 0.048) between groups (Figure 4B). Specifically, the low- and high-intensity interventions prevented the decreased CPM values seen in the sham group (*p* = 0.043 and *p* = 0.025, respectively). There were no significant differences for CPM inter-group comparisons at the elbow and leg (*p* = 0.638 and *p* = 0.577). The procedure of inserting a needle abolished CPM in regions away from the point of intervention, but not in the site of intervention or remote points such as the leg. However, the groups that received PE maintained the ability to produce CPM at all sites tested, regardless of their dosage.

### 3.4. Temporal Summation

Concerning in the elbow, the high-intensity group decreased TS (*p* = 0.014), whereas no significant differences were identified in the other two groups for the intra-group analysis (low-intensity *p* = 0.462; sham *p* = 0.549) (Appendix A). We found significant differences between-groups (*p* = 0.039), with the high-intensity treatment reducing TS compared to sham (*p* = 0.047) (Figure 4C).

Regarding the other measured locations, we found no differences between-groups for TS in the shoulder and leg (*p* = 0.534 and *p* = 0.512, respectively). Nevertheless, some interesting differences appeared in intragroup statistics. The low- and high-intensity groups decreased TS in the leg (*p* = 0.036 and *p* = 0.020) without differences in the shoulder (*p* = 0.283 and *p* = 0.855), while the sham group showed no differences in any of these comparisons (*p* > 0.05).

In summary, high-intensity PE decreased TS in the intervention site (elbow) compared to sham treatment. Although there are intragroup differences pointing out that both PE groups reduced TS in a remote point (leg), we do not have evidence to suggest that this effect is different from the one observed in the sham group.

## 4. Discussion

The primary objective of this study was to investigate the immediate effects on EPM of a single session of two different PE protocols, applied in the common extensor tendon of the lateral epicondyle, compared to a sham intervention. PE increased PPT in the shoulder compared to the sham group, regardless to the intensity applied. In the shoulder, sham group presented an average abolition of the CPM that was preserved in both PE groups. Furthermore, high-intensity PE decreased TS compared to the sham group in the elbow, which was the intervention site. Consequently, a single session of percutaneous electrolysis modulates some aspects of pressure pain perception, in a local and widespread areas, but the results should be interpreted with caution due to the multiplicity of the analyses and the limitations mentioned below. The following paragraphs will discuss the findings in comparison with similar studies.

Our primary outcome (PPT) showed that PE can produce hypoalgesic effects in the shoulder, a segmental-related point to the intervention site (the elbow), suggesting the activation of analgesic mechanisms related to the EPM. Our results are according to previous literature; the remote effects were previously found by Tsai et al. [41] in their study regarding the effects of dry needling in patients with active myofascial trigger points in upper trapezius muscle, suggesting that the neural pathway involved appears to be selectively modulating the spinal cord level corresponding to the area of intervention [21,41]. Contrarily, De-Miguel-Valtierra et al. [9] detected no differences in widespread long-term sensitivity applying PE in patients with shoulder pain. The main justification for these discrepancies could be the inclusion of participants suffering shoulder pain and the follow-up time. Thus, the short-term analgesic mechanisms could be different from long-term effects. On the other hand, no significant differences were found for changes in PPT in the treatment point (elbow) or in a distant non-related point (leg). Other studies involving PE examined the medium and long-term (not immediate) effects of this technique on local PPT, observing an increase in this variable in most of the follow-up time points [6,13,17], which could be due to the inclusion of patients with pain and their consequent long-term improvement in symptomatology and functionality.

Regarding CPM and TS, we should highlight that, as far as we know, this is the first study assessing the effect of PE over them, and PE had no immediate effects for CPM in any of the locations assessed. Schliessbach et al. [42] found similar results in an acupuncture crossover study in healthy subjects, without differences after a single session compared to sham intervention. Lv et al. [43] demonstrated an increase in CPM produced by electro-acupuncture in patients with knee osteoarthritis, but concluded that at least 2 weeks (5 sessions per week) are necessary for this effect to be clinically relevant, which would point out that the number and frequency of sessions is important to generate changes in CPM. Notably, significant differences were detected for CPM in the shoulder between the sham intervention compared to the two PE protocols. Strikingly, the sham PE group reported an almost average abolition of CPM in the shoulder. We do not know how to interpretate this result, as no decline in CPM was reported before with other needling techniques (without current) such as acupuncture [33,42], so further research is advisable.

As far as TS is concerned, a significant decrease in the elbow was found in the high-intensity group compared to the sham group, indicating local effects on TS with a high intensity and short time PE protocol. In a similar direction, Zheng et al. [44] concluded that electro-acupuncture produced a significant decrease in TS in healthy subjects, which lasted for at least 24 h and was mainly segmentally distributed. Otherwise, the trials conducted by Tobbackx et al. [33] and Leite et al. [39] reported no effect on TS of acupuncture and electro-acupuncture treatments in patients with whiplash and chronic low back pain, respectively. On the other hand, although no between-groups differences were shown for TS in the other points assessed, both PE treatment protocols decreased TS in the leg, while no significant differences were found in the sham PE group. Thus, we cannot affirm that PE produces a decrease in this variable in a remote point, but these may be promising results that warrant further research on the effects of PE on widespread TS.

The secondary objective of this clinical trial was to investigate the relation between the effects over the EPM and the dosage of the PE treatment. With the exception that the high-intensity group was the only one to present significant differences compared to the sham group for TS in the elbow, the rest of the outcome showed that the effects were similar between both PE protocols. In fact, no significant differences were found between low- and high-intensity groups. Those results are coherent with the pilot study conducted by Valera-Calero et al. [45], which is the only one to date (to the best of our knowledge) comparing two PE protocols in humans. They found no differences between protocols in sensitivity and pain in patients with patellofemoral pain syndrome. In other studies carried out in rodents, protocols that applied a higher intensity of galvanic current exhibited a greater activation of the molecular mechanisms of tissue inflammatory response and a faster reversal of signs of myofascial trigger points [5,46]. We can summarize that our results and the scarce evidence available seem to indicate that the clinical result from both PE application modalities are similar, thus it would be advisable to use low-intensity protocols as they are perceived to be more comfortable, and this would represent a significant clinical improvement in the application of PE therapy. Nevertheless, the scientific literature on PE application dosage is limited, and more investigations are needed to clarify the time and intensity required in each situation.

It is important to highlight some limitations of the present randomized clinical trial. First, the study was performed in healthy subjects, therefore the findings cannot be extrapolated to daily clinical practice with patients suffering pain, which compromises the external validity of the results. Second, despite the inclusion of a sham procedure and the blinded allocation, the interventions may be perceived as distinct due to their different intensities, which is very difficult to avoid in this type of study. Third, although quantitative sensory testing provides information about the neurophysiological effects of PE, it is impossible for clinical trials to determine with certainty the exact mechanisms of action involved. Fourth, the results of this study are restricted to the immediate effects of assessments. Additionally, only a mechanical pressure stimulus was used during the quantitative sensory testing, so adding other types of stimuli (cutaneous, thermal, or electrical) would be interesting to obtain more information about pain processing. Finally, the discussion of the results is complex due to the shortage of consistent results in the effects of PE on EPM, which are rather variable and small in terms of effect sizes.

The clinical significance of this research resides in the understanding of the mechanisms of PE and the inclusion of the neurophysiological effect in the clinical rationale for the application of this technique. In addition, this study establishes new research lines that would include subjects with chronic pain conditions for observing possible differences with the results obtained in healthy subjects. Other studies would involve a non-intervention control group in order to monitor the effect of time and measurements on outcome variables. Finally, it would be advisable to add other types of stimuli to the pain sensory testing.

## 5. Conclusions

The main conclusion of this study is that an intervention of percutaneous electrolysis applied over the epicondylar common extensor tendon modulates some aspects of pressure pain perception in both local and widespread areas. Secondly, the low- and high-intensity percutaneous electrolysis produce the same hypoalgesia effects with the exception of the temporal summation in the intervention location, indicating that the intensity of PE is probably not related to the endogenous pain modulation. Nevertheless, the interpretation of the observed effects in the different locations is complex and non-straightforward, so further clinical and basic research should be conducted to generate a solid base for percutaneous electrolysis application and optimization.

## Figures and Tables

**Figure 1 jcm-11-02889-f001:**
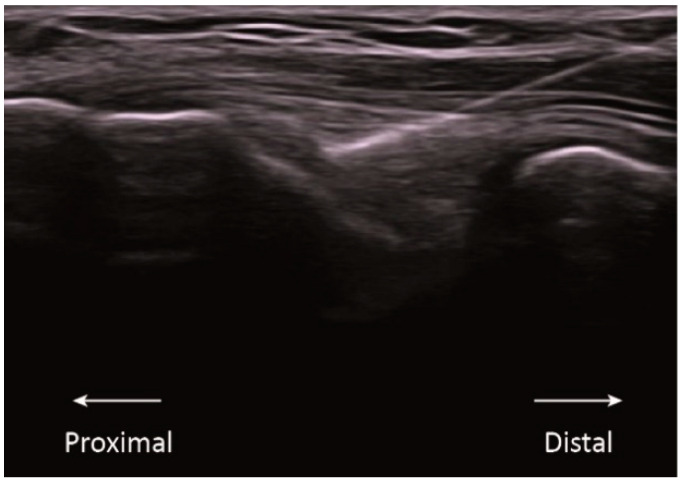
Ultrasound imaging of the application of percutaneous electrolysis in the common extensor tendon of the lateral epicondyle.

**Figure 2 jcm-11-02889-f002:**
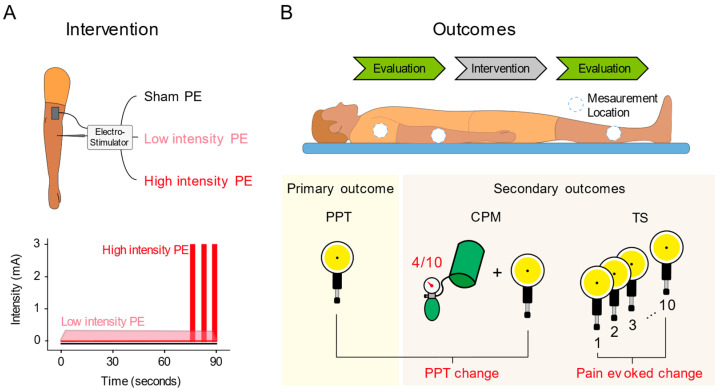
Study methodology illustration. (**A**) The needle intervention, the three PE protocols application, and the representation of the intensity variation through the intervention duration depending of the PE group. (**B**) The outcomes assessment timeline, location, and methodology applied. PPT, Pressure Pain Threshold; CPM, Conditioned Pain Modulation; and TS, Temporal Summation.

**Figure 3 jcm-11-02889-f003:**
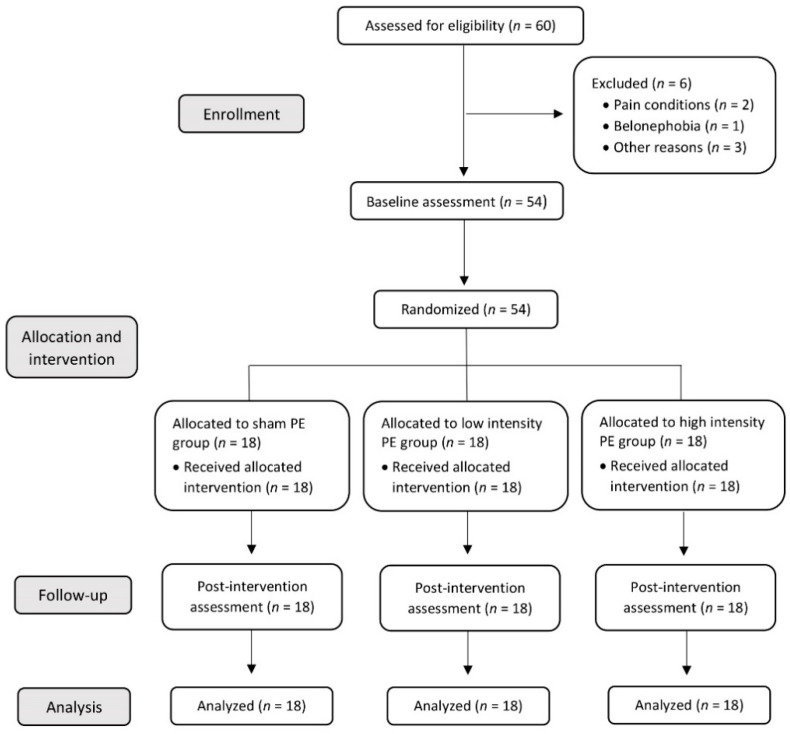
CONSORT flow diagram.

**Figure 4 jcm-11-02889-f004:**
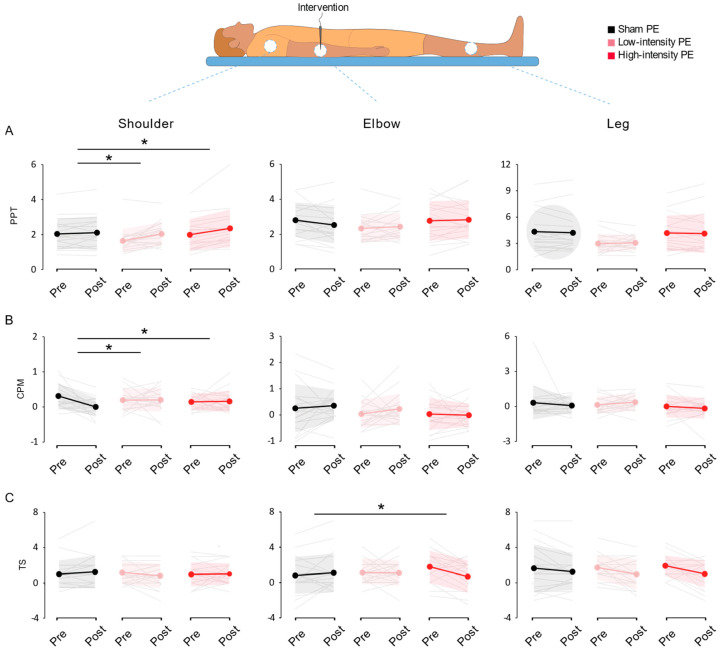
Pressure pain threshold (**A**), conditioned pain modulation (**B**), and temporal summation (**C**) measured in the three locations assessed. Individual data are presented as grey lines, and colored line represents mean ± SD. Asterisks indicates significations for between-group comparisons (*p* < 0.05; Bonferroni post hoc test or Mann–Whitney *U* test).

**Table 1 jcm-11-02889-t001:** Baseline demographic and outcome measures.

Variable	Total (*n* = 54)	Sham (*n* = 18)	Low-Intensity (*n* = 18)	High-Intensity (*n* = 18)	*p*
**Demographic**					
Age (years)	22.96 (3.63)	23.50 (3.09)	23.33 (4.47)	22.06 (3.19)	0.26
Weight (kg)	67.17 (10.45)	67.93 (11.38)	66.66 (11.19)	66.92 (9.22)	0.55
Height (m)	1.73 (0.08)	1.74 (0.07)	1.76 (0.09)	1.74 (0.95)	0.75
BMI (kg/m^2^)	22.23 (2.21)	22.31 (3.07)	22.46 (1.92)	21.91 (1.43)	0.43
IPAQ-SF (METs-min-week)	2668 (2687)	3249 (3446)	2709 (2572)	2045 (1804)	0.54
Gender (male)	34 (63.0)	12 (66.6)	9 (50.0)	13 (72.2)	0.36
Smoking (yes)	8 (14.8)	3 (16.7)	2 (11.1)	3 (16.7)	0.86
**Baseline Outcomes**					
PPT elbow (kg/cm^2^)	2.62 (0.97)	2.81 (0.96)	2.32 (0.77)	2.74 (1.12)	0.30
PPT shoulder (kg/cm^2^)	1.88 (0.81)	2.04 (0.86)	1.62 (0.66)	1.97 (0.88)	0.18
PPT leg (kg/cm^2^)	3.82 (1.88)	4.35 (2.20)	2.96 (1.07)	4.14 (1.96)	0.08
CPM elbow (kg/cm^2^)	0.11 (0.69)	0.26 (0.92)	0.03 (0.51)	0.03 (0.60)	0.56
CPM shoulder (kg/cm^2^)	0.22 (0.30)	0.31 (0.35)	0.19 (0.30)	0.15 (0.22)	0.22
CPM leg (kg/cm^2^)	0.21 (0.30)	0.32 (1.44)	0.13 (0.72)	0.01 (0.94)	0.85
TS elbow (0–10)	1.23 (1.76)	0.81 (2.07)	1.11 (1.23)	1.78 (1.82)	0.24
TS shoulder (0–10)	1.04 (1.33)	1.00 (1.53)	1.17 (1.15)	0.97 (1.36)	0.82
TS leg (0–10)	1.75 (1.83)	1.64 (2.69)	1.69 (1.33)	1.92 (1.15)	0.89

Data are expressed as Mean (SD) for quantitative variables, or in frequencies (%) for qualitative variables. BMI indicates Body Mass Index; IPAQ-SF, short version of the International Physical Activity Questionnaire; PPT, Pressure Pain Threshold; CPM, Conditioned Pain Modulation; and TS, Temporal Summation. *p*-values correspond to One-Way ANOVA test (or Kruskal–Wallis test) for quantitative variables and to Chi-squared test for qualitative variables.

**Table 2 jcm-11-02889-t002:** Intra- and between-groups differences in outcome measures.

Measure	Group	Pre	Post	Intra-Group Differences	Between-Group Differences
**Pressure Pain Threshold (PPT)**					
**PPT elbow (kg/cm^2^)**	Sham	2.81 (0.96)	2.53 (1.05)	−0.28 (−0.58 to 0.03); *d* = 0.28	Sham vs. Low −0.38 (−0.91 to 0.15); *d* = 0.60
	Low	2.32 (0.77)	2.42 (0.75)	0.10 (−0.22 to 0.42); *d* = 0.13	Sham vs. High −0.34 (−0.86 to 0.19); *d* = 0.52
	High	2.74 (1.12)	2.80 (1.08)	0.06 (−0.27 to 0.39); *d* = 0.05	Low vs. High 0.04 (−0.49 to 0.57); *d* = 0.06
**PPT shoulder (kg/cm^2^)**	Sham	2.04 (0.86)	2.11 (0.67)	0.07 (−0.08 to 0.23); *d* = 0.08	**Sham vs. Low −0.32 (−0.67 to 0.03); *d* = 0.74 ***
	Low	1.62 (0.66)	2.01 (0.67)	**0.39 (0.13 to 0.66); *d* = 0.59 ****	**Sham vs. High −0.29 (−0.64 to 0.06); *d* = 0.80 ***
	High	1.97 (0.88)	2.33 (1.08)	**0.36 (0.16 to 0.56); *d* = 0.37 ****	Low vs. High 0.03 (−0.32 to 0.38); *d* = 0.07
**PPT leg (kg/cm^2^)**	Sham	4.35 (2.20)	4.22 (2.20)	−0.13 (−0.50 to 0.24); *d* = 0.06	Sham vs. Low −0.20 (−0.49 to 0.90); *d* = 0.25
	Low	2.96 (1.07)	3.03 (0.84)	0.07 (−0.38 to 0.54); *d* = 0.08	Sham vs. High −0.06 (−0.63 to 0.76); *d* = 0.09
	High	4.14 (1.96)	4.07 (2.19)	−0.07 (−0.48 to 0.36); *d* = 0.03	Low vs. High 0.14 (−0.56 to 0.83); *d* = 0.16
**Conditioned Pain Modulation (CPM)**					
**CPM elbow (kg/cm^2^)**	Sham	0.25 (0.92)	0.35 (0.59)	0.10 (−0.29 to 0.49); *d* = 0.13	Sham vs. Low −0.09 (−0.70 to 0.52); *d* = 0.12
	Low	0.03 (0.51)	0.22 (0.58)	0.19 (−0.18 to 0.56); *d* = 0.35	Sham vs. High 0.14 (−0.47 to 0.75); *d* = 0.19
	High	0.03 (0.60)	−0.01 (0.44)	−0.04 (−0.38 to 0.30); *d* = 0.08	Low vs. High 0.23 (−0.38 to 0.84); *d* = 0.33
**CPM shoulder (kg/cm^2^)**	Sham	0.31 (0.35)	0.00 (0.26)	**−0.31 (−0.52 to −0.10); *d* = 1.00 ****	**Sham vs. Low −0.32 (−0.66 to 0.03) *d* = 0.72 ***
	Low	0.19 (0.30)	0.20 (0.33)	0.01 (−0.22 to 0.24); *d* = 0.02	**Sham vs. High −0.32 (−0.67 to 0.02); *d* = 0.82 ***
	High	0.15 (0.22)	0.16 (0.01)	0.01 (−0.17 to 0.20); *d* = 0.05	Low vs. High 0.00 (−0.35 to −0.34); *d* = 0.13
**CPM leg (kg/cm^2^)**	Sham	0.32 (1.44)	0.08 (0.64)	−0.24 (−1.10 to 0.61); *d* = 0.22	Sham vs. Low −0.47 (−1.51 to 0.57); *d* = 0.35
	Low	0.13 (0.72)	0.36 (0.69)	0.23 (−0.19 to 0.64); *d* = 0.32	Sham vs. High −0.07 (−1.12 to 0.97); *d* = 0.05
	High	0.01 (0.94)	−0.16 (0.87)	−0.17 (−0.70 to 0.36); *d* = 0.19	Low vs. High 0.40 (−0.65 to 1.44); *d* = 0.41
**Temporal Summation (TS)**					
**TS elbow** **(0–10)**	Sham	0.81 (2.07)	1.11 (2.17)	0.30 (−0.48 to 1.09); *d* = 0.14	Sham vs. Low 0.33 (−1.02 to 1.69); *d* = 0.21
	Low	1.11 (1.23)	1.08 (1.37)	−0.03 (−0.86 to 0.80); *d* = 0.02	**Sham vs. High 1.41 (0.06 to 2.77); *d* = 0.87 ***
	High	1.78 (1.82)	0.67 (1.83)	**−1.11 (−1.94 to −0.28); *d* = 0.61 ***	Low vs. High 1.08 (−0.28 to −0.06); *d* = 0.65
**TS shoulder (0–10)**	Sham	1.00 (1.53)	1.25 (1.86)	0.25 (−0.16 to 0.66); *d* = 0.15	Sham vs. Low 0.61 (−0.37 to 1.59); *d* = 0.54
	Low	1.17 (1.15)	0.81 (1.35)	−0.36 (−1.05 to 0.33); *d* = 0.29	Sham vs. High 0.19 (−0.78 to 1.17); *d* = 0.18
	High	0.97 (1.36)	1.03 (1.17)	0.06 (−0.58 to 0.69); *d* = 0.04	Low vs. High −0.42 (−1.39 to 0.56); *d* = 0.31
**TS leg** **(0–10)**	Sham	1.64 (2.69)	1.25 (2.09)	−0.39 (−1.04 to 0.26); *d* = 0.16	Sham vs. Low 0.36 (−0.79 to 1.51); *d* = 0.27
	Low	1.69 (1.33)	0.94 (1.65)	**−0.75 (−1.44 to −0.06); *d* = 0.50 ***	Sham vs. High 0.53 (−0.62 to 1.67); *d* = 0.38
	High	1.92 (1.15)	1.00 (1.51)	**−0.92 (−1.65 to −0.19); *d* = 0.68 ***	Low vs. High 0.17 (−0.98 to 1.31); *d* = 0.12

Data are expressed as Mean (SD) for pre- and post-intervention measures and as Mean (CI) for mean differences. * *p* < 0.05; ** *p* < 0.01. Significations correspond to paired *t*-test (or Wilcoxon test) for intra-group differences and to Bonferroni post hoc test (or Mann–Whitney *U* test) for between-group differences.

## Data Availability

Data are held securely by the research team and may be available upon reasonable request and with relevant approvals in place.

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
