# Peer review of "Endogenous Pain Modulation in Response to a Single Session of Percutaneous Electrolysis in Healthy Population: A Double-Blinded Randomized Clinical Trial"

_jcm, 2022, doi:10.3390/jcm11102889_

Round 1

Reviewer 1 Report

The manuscript by Varela-Rodriguez et al. investigated the effects of percutaneous electrolysis at two intensities on three quantitative sensory testing paradigms.  The authors placed a needle electrode at the lateral epicondyle and delivered sham, low intensity, or high-intensity stimulation at a total intensity of 27 mC.  The primary outcome was pain pressure thresholds at the elbow, an adjacent site (shoulder), and a distal site (leg) before and after the stimulation.  Secondary outcomes included condition pain modulation (to blood pressure cuff) and temporal summation to ten consecutive stimuli, delivered at the same sites.  The patients were healthy volunteers without a history of chronic pain, and all enrolled patients were able to complete the study.  The authors found that PE at both intensities increased pain pressure thresholds at the shoulder whereas sham had no effect. PE did not affect PPT at the elbow and the leg, suggesting that PE may affect segmental mechanisms, but not local and distal ones.  PE only had an effect on CPM in the sham treatment, suggesting that one treatment is insufficient to increase CPM -despite other studies which have shown that PE can modulate CPM in patients with chronic pain.  PE at high intensity decreased temporal summation at the elbow and leg, whereas low intensity only decreased TS at the leg.  There was no difference between high and low-intensity stimulation, suggesting that future studies can use low stimulation which is more comfortable for patients.

This is a well-designed and conducted clinical study that will be of interest to the pain neuroscience community as it provides a physiological basis for some of the analgesic effects observed with PE.  I wonder if some of the observed effects are simply due to statistical variability given the small sample size and multiple comparisons.  That being said, the authors state that they used ANOVA with Bonferroni correction which is appropriate.

I only have some minor issues that need to be addressed.

  • Line 7: Nursery should say Nursing.
  • Line 138: Coulomb’s law is used inappropriately in this context.
  • Figure 1 needs orientation arrows (medial – lateral, proximal - distal) to make it easier to interpret.

Author Response

The manuscript by Varela-Rodriguez et al. investigated the effects of percutaneous electrolysis at two intensities on three quantitative sensory testing paradigms.  The authors placed a needle electrode at the lateral epicondyle and delivered sham, low intensity, or high-intensity stimulation at a total intensity of 27 mC.  The primary outcome was pain pressure thresholds at the elbow, an adjacent site (shoulder), and a distal site (leg) before and after the stimulation.  Secondary outcomes included condition pain modulation (to blood pressure cuff) and temporal summation to ten consecutive stimuli, delivered at the same sites.  The patients were healthy volunteers without a history of chronic pain, and all enrolled patients were able to complete the study.  The authors found that PE at both intensities increased pain pressure thresholds at the shoulder whereas sham had no effect. PE did not affect PPT at the elbow and the leg, suggesting that PE may affect segmental mechanisms, but not local and distal ones.  PE only had an effect on CPM in the sham treatment, suggesting that one treatment is insufficient to increase CPM -despite other studies which have shown that PE can modulate CPM in patients with chronic pain.  PE at high intensity decreased temporal summation at the elbow and leg, whereas low intensity only decreased TS at the leg.  There was no difference between high and low-intensity stimulation, suggesting that future studies can use low stimulation which is more comfortable for patients.

This is a well-designed and conducted clinical study that will be of interest to the pain neuroscience community as it provides a physiological basis for some of the analgesic effects observed with PE.  I wonder if some of the observed effects are simply due to statistical variability given the small sample size and multiple comparisons.  That being said, the authors state that they used ANOVA with Bonferroni correction which is appropriate.

I only have some minor issues that need to be addressed.

First of all, we deeply appreciate all the suggestions of the reviewer 1.

  • Line 7: Nursery should say Nursing.
    • We have addressed it.
  • Line 138: Coulomb’s law is used inappropriately in this context.
    • We have eliminated the Coulomb’s law reference.
  • Figure 1 needs orientation arrows (medial – lateral, proximal - distal) to make it easier to interpret.
    • We have modified Figure 1 as the reviewer suggested, including a spatial orientation guide (proximal-distal arrows).

Reviewer 2 Report

Reviewer Comments

Thank you very much for the opportunity to review the manuscript submission entitled: Endogenous Pain Modulation in Response to a Single Session of Percutaneous Electrolysis: A Double-Blinded Randomized Clinical Trial.

The current paper aims to demonstrate the aims to investigate 12 whether percutaneous electrolysis (PE) is able to activate endogenous pain modulation and 13 whether its effects are dependent on the dosage of the galvanic current. The data is interesting, and it has a relevant rationale, however, some limitations and constructive comments are pointed below:

General comments:

How is this study different in terms of underlying mechanisms on pain  and other mechanics compared to previous studies that are published which are mentioned below.

  • Rodríguez-Huguet, M., Góngora-Rodríguez, J., Lomas-Vega, R., Martín-Valero, R., Díaz-Fernández, Á., Obrero-Gaitán, E., Ibáñez-Vera, A.J. and Rodríguez-Almagro, D., 2020. Percutaneous electrolysis in the treatment of lateral epicondylalgia: A single-blind randomized controlled trial. Journal of Clinical Medicine9(7), p.2068.
  • Rodríguez-Huguet, M., Góngora-Rodríguez, J., Rodríguez-Huguet, P., Ibañez-Vera, A.J., Rodríguez-Almagro, D., Martín-Valero, R., Díaz-Fernández, Á. and Lomas-Vega, R., 2020. Effectiveness of percutaneous electrolysis in supraspinatus tendinopathy: A single-blinded randomized controlled trial. Journal of Clinical Medicine9(6), p.1837.
  • Gómez-Chiguano, G.F., Navarro-Santana, M.J., Cleland, J.A., Arias-Buría, J.L., Fernández-de-Las-Peñas, C., Ortega-Santiago, R. and Plaza-Manzano, G., 2021. Effectiveness of ultrasound-guided percutaneous electrolysis for musculoskeletal pain: A systematic review and meta-analysis. Pain Medicine22(5), pp.1055-1071.

Specific comments

Title and Abstract

  • It is advisable to mention the type of sample you investigated in the title.
  • Include MeSH terms as keywords.

Introduction

  • The scientific background and rationale for the investigation need to be emphasized with the current literature.
  • The hypotheses of the study need to be stated.

Statistical methods and results

  • Compute minimal clinically important difference (MCID) is the smallest change in a treatment outcome that an individual patient would identify as important and which would indicate a change in the patient's management.

Discussion

  • Give a cautious overall interpretation of results considering objectives, limitations, the multiplicity of analyses, and results from similar studies.
  • Discuss the generalizability (external validity) of the study results.
  • Emphasize clinical significance and scope for future research.

Author Response

Reviewer Comments

Thank you very much for the opportunity to review the manuscript submission entitled: Endogenous Pain Modulation in Response to a Single Session of Percutaneous Electrolysis: A Double-Blinded Randomized Clinical Trial.

The current paper aims to demonstrate the aims to investigate 12 whether percutaneous electrolysis (PE) is able to activate endogenous pain modulation and 13 whether its effects are dependent on the dosage of the galvanic current. The data is interesting, and it has a relevant rationale, however, some limitations and constructive comments are pointed below:

First of all, we deeply appreciate all the suggestions of the reviewer 2.

General comments:

How is this study different in terms of underlying mechanisms on pain  and other mechanics compared to previous studies that are published which are mentioned below.

  • Rodríguez-Huguet, M., Góngora-Rodríguez, J., Lomas-Vega, R., Martín-Valero, R., Díaz-Fernández, Á., Obrero-Gaitán, E., Ibáñez-Vera, A.J. and Rodríguez-Almagro, D., 2020. Percutaneous electrolysis in the treatment of lateral epicondylalgia: A single-blind randomized controlled trial. Journal of Clinical Medicine9(7), p.2068.
  • Rodríguez-Huguet, M., Góngora-Rodríguez, J., Rodríguez-Huguet, P., Ibañez-Vera, A.J., Rodríguez-Almagro, D., Martín-Valero, R., Díaz-Fernández, Á. and Lomas-Vega, R., 2020. Effectiveness of percutaneous electrolysis in supraspinatus tendinopathy: A single-blinded randomized controlled trial. Journal of Clinical Medicine9(6), p.1837.
  • Gómez-Chiguano, G.F., Navarro-Santana, M.J., Cleland, J.A., Arias-Buría, J.L., Fernández-de-Las-Peñas, C., Ortega-Santiago, R. and Plaza-Manzano, G., 2021. Effectiveness of ultrasound-guided percutaneous electrolysis for musculoskeletal pain: A systematic review and meta-analysis. Pain Medicine22(5), pp.1055-1071.

The first two publications mentioned above state that the underlying mechanisms of the percutaneous electrolysis technique are focused in the posible electrolysis reaction, which generates a controlled inflammatory response that facilitates the repair of the soft tissues. Therefore, they follow the classical path of limiting the effects of the technique to a mechanical and biochemical paradigm, meanwhile our study focusses on neurophysiological effects. Finally, the meta-analysis mentioned in third place is one of the first publications on percutaneous electrolysis in which the possible neurophysiological effect of the technique is discussed and it is the field on which our study has been focused. The difference between our work and this meta-analysis is that our work is a clinical trial that provides more evidence about the percutaneous electrolysis effects with high quality methodology, that it is a common limitation of systematic review and meta-analysis, the poor quality of the clinical trials.

Specific comments

Title and Abstract

  • It is advisable to mention the type of sample you investigated in the title.
    • We agree with the reviewer and we have mentioned in the title that our work is performed in “healthy population”.
  • Include MeSH terms as keywords.
    • We have addressed it.

Introduction

  • The scientific background and rationale for the investigation need to be emphasized with the current literature.
    • We have added more updated references, but they are not reflected in the “Track changes” of MS Word (references numbers are 13, 17, 24, 27, in lines 46, 55, 67 and 74, respectively).
  • The hypotheses of the study need to be stated.
    • We have stated the hypotheses at the end of the introduction, line 86: Thus, we first hypothesize that a single session of percutaneous electrolysis will be able to activate EPM in a greater manner than the application of a sham intervention. Our second hypothesis is that activation of EPM will be higher with the high-intensity protocol than with the low-intensity protocol.

Statistical methods and results

  • Compute minimal clinically important difference (MCID) is the smallest change in a treatment outcome that an individual patient would identify as important and which would indicate a change in the patient's management.
    • Thank you very much for your comment. It would be very interesting to calculate MCID, but one of the difficulties of this research is that the variables related to the endogenous pain modulation are hardly related to the possible clinical change that we obtain in patients.

Discussion

  • Give a cautious overall interpretation of results considering objectives, limitations, the multiplicity of analyses, and results from similar studies.
    • We have written a cautious overall interpretation in line 337: Consequently, a single session of percutaneous electrolysis modulates some aspects of pressure pain perception, in a local and widespread areas, but the results should be interpreted with caution due to the multiplicity of the analyses and the limitations mentioned below. The following paragraphs will discuss the findings in comparison with similar studies.
  • Discuss the generalizability (external validity) of the study results.
    • We have indicated that the limitation of involving healthy humans, “compromises the external validity of the results”, at line 406.
  • Emphasize clinical significance and scope for future research.
    • We have indicated the clinical significance and the scope for future research at the end of the discussion, line 418: The clinical significance of this research resides in the understanding of the mechanisms of PE and the inclusion of the neurophysiological effect in the clinical rationale for the application of this technique. In addition, this study establishes new research lines that would include subjects with chronic pain conditions for observing possible differences with the results obtained in healthy subjects. Other studies would involve a non-intervention control group in order to monitor the effect of time and measurements on outcome variables. Finally, it would be advisable to add other types of stimuli to the pain sensory testing.”

Reviewer 3 Report

Review of manuscript Endogenous Pain Modulation in Response to a Single Session
of Percutaneous Electrolysis: A Double-Blinded Randomized Clinical Trial.

Thank you for possibility to review this interesting, well- written manuscript.

The research regards Percutaneous electrolysis - ultrasound-guided minimally invasive technique consisting of the application of a galvanic electrolytic current through an acupuncture needle placed into affected soft tissue

The primary aim of this randomized clinical trial was to investigate if a single session of percutaneous electrolysis activates endogenous pain modulation mechanisms compared to a sham intervention in asymptomatic healthy participants. The secondary aim was to determine if these potential effects on EPM are different between the application of two different protocols of PE (low intensity galvanic current during longer time or high intensity during shorter time).

The protocol of the clinical trial received approval from the Ethics Committee. The reporting of this study was conducted according to the CONSORT 2010 Statement. The clinical trial was registered NCT05097937.

Tables are clear and easy to follow. Discussion is well written. Conclusions are based on results.

Author Response

Review of manuscript Endogenous Pain Modulation in Response to a Single Session
of Percutaneous Electrolysis: A Double-Blinded Randomized Clinical Trial.

Thank you for possibility to review this interesting, well- written manuscript.

The research regards Percutaneous electrolysis - ultrasound-guided minimally invasive technique consisting of the application of a galvanic electrolytic current through an acupuncture needle placed into affected soft tissue

The primary aim of this randomized clinical trial was to investigate if a single session of percutaneous electrolysis activates endogenous pain modulation mechanisms compared to a sham intervention in asymptomatic healthy participants. The secondary aim was to determine if these potential effects on EPM are different between the application of two different protocols of PE (low intensity galvanic current during longer time or high intensity during shorter time).

The protocol of the clinical trial received approval from the Ethics Committee. The reporting of this study was conducted according to the CONSORT 2010 Statement. The clinical trial was registered NCT05097937.

Tables are clear and easy to follow. Discussion is well written. Conclusions are based on results.

Thank you very much on behalf of all the authors for your comments.